materials science/chemical engineering

butanol, pervaporation, ionic liquid, MCM-41, membrane

**Author for correspondence:**
Yanhui Wu
e-mail: wuyanhui@tongji.edu.cn

This article has been edited by the Royal Society of Chemistry, including the commissioning, peer review process and editorial aspects up to the point of acceptance.

# Ionic liquid-modified MCM-41-polymer mixed matrix membrane for butanol pervaporation

## Yifang Li[1], Dandan Yan[2] and Yanhui Wu[2]

[1]Shanghai Shenglan Petrochemical Engineering Technology Co. Ltd, Shanghai 201200, People's Republic of China
[2]Shanghai Key Laboratory of Chemical Assessment and Sustainability, School of Chemical Science and Engineering, Tongji University, Shanghai 200092, People's Republic of China

YW, 0000-0003-4029-7585

Because of the preferential butanol selectivity of some ionic liquids (ILs), an increasing amount of research has appeared regarding their application in butanol separation. In this research, two ionic liquids, namely, 1-ethyl-3-vinylimidazolium bis[(trifluoromethyl)sulfonyl]imide ([EVIM][Tf$_2$N], IL1) and *N*-octyl-pyridinium bis[(trifluoromethyl)sulfonyl]imide ([OMPY] [Tf$_2$N], IL2), were applied to modify a mesoporous molecular sieve MCM-41. The IL-modified MCM-41 samples were characterized by XPS, BET, XRD, SEM and TEM. The ionic liquid-modified MCM-41 was incorporated into the polymer PEBA to prepare mixed matrix membranes to study the influences of the filling of IL-modified MCM-41 and operating conditions on the performance of the mixed matrix membrane for butanol pervaporation. The results indicated that the pervaporation performance of the PEBA membrane was enhanced by the incorporation of IL-modified MCM-41. When the temperature of the feeding liquid was 35°C and the mass fraction of butanol was 2.5 wt%, the 5% MCM-41-IL2-PEBA membrane showed a permeation flux of 421.7 g m$^{-2}$ h$^{-1}$ and a separation factor of 25.4. The permeation flux and the separation factor of the membrane increased as the temperature of the feeding liquid increased. The results of the long-period experiment suggested that the 5% MCM-41-IL2-PEBA membrane exhibited high stability within 100 h of operation.

## 1. Introduction

As a novel biomass fuel, bio-butanol has a better energy density and performance than bioethanol [1]. Bio-butanol can be conveniently blended with gasoline in any ratio. In addition, butanol can be transported using the existing petroleum

pipeline because of the low pressure of butanol vapour and its non-corrosive property. With the growing shortage of fossil energy and the rapid development of bio-technology, biomass energy produced by biological fermentation has drawn more and more research attention [2].

Clostridium acetobutylicum is commonly used in bio-fermentation to produce butanol, in which the fermentation products also include some by-products, such as acetone and ethanol. As a result, it is also known as ABE fermentation. In the industrial fermentation of butanol, the production basically stops when the mass concentration of butanol reaches $10–16 \, g \, l^{-1}$, because butanol has an inhibition effect at levels greater than this value [3]. To enhance butanol yield and reduce the production cost, it is important to continuously separate butanol from the fermentation broth, thereby reducing the butanol concentration in the fermentation broth and maintaining the activity of clostridium cells [4].

There are a variety of methods for butanol separation, such as gas stripping, liquid–liquid extraction, rectification, adsorption and pervaporation (PV). PV is non-polluting, consumes a moderate amount of energy and requires a smaller investment, which are suitable characteristics for the separation of a small amount of organic matter from aqueous solutions. The PV technique has already been researched for separating butanol, acetone and ethanol from butanol fermentation liquid. During the process, the solvent components permeate through the pervaporation membrane, retaining the broth.

Poly(ether-block-amide) (PEBA) is an organophilic copolymer that has been extensively studied when applied to organic permselective pervaporation because of its excellent mechanical performance, thermal stability and chemical stability [5,6]. With a high polyether content, favourable flexibility of the polymer segment, good organophilic properties and excellent film-forming properties, PEBA is widely used in studies on organic matter separation [6–8]. A PEBA membrane was used for the PV separation of an ABE solution by Yen et al. [9], who reported that the PEBA membrane had high butanol selectivity. Polymer materials use relatively simple preparation processes to form membranes. However, they also have poor anti-swelling performance, and a trade-off must usually be made when using a polymer membrane [10]. The concept of organic–inorganic hybrid membranes was proposed in 1986. Since then, much research has been devoted to the study of the combination of organic polymer membranes and inorganic particles. Researchers have mechanically blended inorganic and organic phases so that the two materials were combined by interactions, such as hydrogen bonds and van der Waals forces. The added inorganic porous materials for the separation of organic compounds are usually hydrophobic materials including Silicalite-1 [11–13], ZSM-5 [14–16], carbon molecular sieve (CMS) [17], carbon nano tubes (CNTs) [18,19], carbon black [5,20] and MOFs [21]. These compounds can preferentially adsorb organic molecules or provide diffusion channels to improve the selectivity or permeation flux because of the good hydrophobic effect, the high specific surface area and pore volume. Liu et al. [21] filled ZIF-71 into PEBA to prepare mixed matrix membranes (MMMs) for PV separation of the ABE solution, showing that the separation performance of a ZIF-71-modified PEBA membrane can be improved. When the filling amount of ZIF-71 was 20 wt% and the feeding liquid temperature was 37°C, MMMs showed a permeation flux of $520.2 \, g \, m^{-2} \, h^{-1}$ and a separation factor of 18.8.

In addition to the above-mentioned filling materials, mesoporous materials have been attracting interest because of their regular pore structure and large pore volume [22]. MCM-41 is the most common mesoporous filling material, with pore channels that can provide enough space for molecular adsorption and diffusion. The Si–OH groups on its surface can form hydrogen bonds with some molecules to enhance the adsorption. In addition, polymer chain segments can enter the mesopores and increase the adhesion between organic and inorganic materials [23].

Compared with traditional solvents, ionic liquid (IL), a type of organic salt, offers many advantages, such as low vapour pressure, excellent dissolving capacity, favourable thermal stability and property tunability [24]. Therefore, it has received extensive attention over the past few years. IL has been used directly for butanol extraction by some researchers based on the high solubility of butanol. In a study by Garcia et al. [25], butanol was extracted using a non-fluorinated IL. They found that IL [TOAM][Naph] exhibited a very high distribution coefficient (21) and selectivity (274). In addition, the energy consumption of butanol extraction with [TOAM][Naph] could be reduced by 73% compared with conventional distillation. In a study by Ha et al. [26], 11 imidazole ILs were used to extract butanol solution, the result of which suggested that the distribution of butanol between water and IL was primarily determined by the hydrophobicity of the anions in the IL. The sequence of the anions of the IL in descending order of hydrophobicity was $[Tf_2N]^- > [Pf_6]^- > [Bf_4]^- > [TfO]^-$. When IL was used for the PV separation of the alcohol–water solution, most studies were concentrated on fixing IL into the polymer membrane [27]. Cascon et al. [28] prepared a gel-IL composite membrane for the PV separation of a butanol solution by gelling ionic liquid with PVDF-co-HFP and then dipping the gel into the porous support membrane. PVDF-co-HFP blending with ionic liquid effectively improved

the pervaporation performance. When the mass ratio of PVDF-co-HFP to $[P_{6,6,6,14}][dca]$ was 1:2.5, the butanol permeation flux and separation factor of the ionic liquid gel-based membrane were 226 g m$^{-2}$ h$^{-1}$ and 68, respectively.

Although IL offers significant advantages over traditional organic solvents, there are some disadvantages, such as high viscosity and recycling difficulty. In addition, the use of IL is also faced with industrial problems, such as high costs, resulting from the large consumption of expensive materials. Therefore, it is useful to immobilize IL on some supports [28,29]. Connecting IL to a carrier such as a molecular sieve is a common immobilizing method. After immobilizing the ionic liquid, the molecular sieves were functionalized or modified, sharing the advantages of a molecular sieve and the ionic liquid simultaneously.

The mesoporous molecular sieve MCM-41 has a high specific surface and pore volume, and ionic liquids containing the anion [Tf$_2$N]$^-$ have good solubility for butanol. To improve the permeation flux and separation factor of the pervaporation membrane for butanol aqueous solutions simultaneously, in this study, two ionic liquids with the hydrophobic anion [Tf$_2$N]$^-$ were selected to couple with a MCM-41 mesoporous sieve to prepare IL-modified MCM-41. Then, the IL-modified MCM-41 was incorporated into the PEBA membrane to prepare a series of mixed matrix membranes. The IL-modified MCM-41 was characterized with XPS, BET, XRD, SEM and TEM. The morphology, thermal stability and swelling behaviour of the mixed matrix membranes were analysed. Afterwards, pervaporation experiments were conducted to evaluate the membrane performance and stability of the mixed matrix membranes.

# 2. Experimental method

## 2.1. Materials

The 1-ethyl-3-vinylimidazolium bis [(trifluoromethyl)sulfonyl]-imide ([EVIM][Tf$_2$N], IL1), with a purity of 99%, and N-octyl-pyridinium bis((trifluoromethyl)sulfonyl)imide ([OMPY][Tf$_2$N], IL2), with a purity of 99%, were produced by the Shanghai Cheng Jie Chemical Co. LTD. The 3-chloropropyltrimethoxysilane, with a purity of 98%, was supplied by the Aladdin Group. MCM-41 zeolite with a particle diameter of 1.2 μm was produced by Nankai University Catalyst Co. Ltd. Nitric acid, n-hexane, concentrated hydrochloric acid, toluene, methanol, acetonitrile, triethylamine and ethyl acetate, with analytic purity, were provided by the Sinopharm Chemical Reagent Co. Ltd. PEBA 2533 was produced by the Arkema Group.

## 2.2. Preparation of the IL-modified MCM-41

An appropriate amount of MCM-41 was weighed and added to the nitrate solution. The solution was stirred at room temperature for 24 h, followed by washing with distilled water and ethanol. The washed specimen was then dried at 120°C for 12 h (to activate the molecular sieve) [30]. A quantity of 2 g activated MCM-41 was weighed and placed in a 250 ml three-necked flask. Next, 100 ml of toluene (treated by refluxing with metal sodium followed by distillation), 5 ml of 3-chloropropyltrimethoxysilane and 2 ml of triethylamine (as the catalyst) were added to the flask, which was then mixed by magnetic stirring at 80°C for 24 h. After the reaction, the mixture was cooled to room temperature and was vacuum filtered. Then, the product was washed using toluene, methanol, a water/methanol mixture (1:1), distilled water and methanol to obtain a white solid powder. The powder was dried in a vacuum at 80°C for 8 h. Subsequently, the silanized molecular sieve, MCM-41-Si, was produced.

The prepared MCM-41-Si was placed into a 250-ml three-necked flask, with 100 ml acetonitrile added as a solvent. A quantity of 0.005 mol ionic liquid (IL) was also added to the flask. The mixture received magnetic stirring at 80°C for 24 h. When the reaction was completed, the mixture was cooled to room temperature and was vacuum filtered. The product was washed using ethyl acetate, 0.1 mol l$^{-1}$ hydrochloric acid, distilled water and methanol, sequentially. The resulting white solid was dried in an oven at 80°C for 12 h. The modified molecular sieves were labelled MCM-41-IL1 and MCM-41-IL2, corresponding to whether ionic liquid [EVIM][Tf$_2$N] or [OMPY][Tf$_2$N] were used. Figure 1 shows the preparation routes of ionic liquid-modified MCM-41. MCM-41 was first silanized by 3-chloropropyltriethoxysilane. Because the anion [Tf2N]$^-$ of the two ionic liquids is a good nucleophile, the nucleophilic substitution reaction can take place between IL1/IL2 and the silanized MCM-41 [31,32]. At the same time, IL1 or IL2 can also couple with MCM-41 by hydrogen bond.

**Figure 1.** Preparation scheme of ionic liquid-modified MCM-41 and the structure of IL1 and IL2.

## 2.3. Preparation of the mixed matrix membranes

The IL-modified MCM-41, as described in §2.2, was added to butanol. The solution received ultrasonic dispersion for 30 min. PEBA granules were added to the butanol, which was magnetically stirred in a water bath at a constant temperature of 70°C for 1 h. Subsequently, the ultrasonic dispersed molecular sieve was added to the PEBA solution. The solution was stirred in a water bath at 70°C for another 2 h until the membrane casting solution was evenly mixed. It was then left standing for defoaming. When the membrane casting solution was cooled to 40°C, it was poured onto a horizontal glass plate. The membrane was scraped using a glass slicker and kept at room temperature for a period of time. When most of the solvent volatilized, the membrane was moved into an oven and dried at 60°C. The membrane was not peeled off until the solvent had been completely volatilized. The membranes modified with IL-modified MCM-41 mass fractions of 2 wt%, 5 wt% and 10 wt% were denoted as 2% MCM-41-IL-PEBA, 5% MCM-41-IL-PEBA and 10% MCM-41-IL-PEBA, respectively.

## 2.4. Characterization

X-ray photoelectron spectroscopy (XPS, AXIS Ultra, Shimadzu-Kratos) was used to determine the types and quantities of elements in pristine MCM-41 and IL-modified MCM-41. X-ray radiation (200 W, 1253 eV) was used as the source of the excitation. The crystal structure of pristine MCM-41 and IL-modified MCM-41 were measured by X-ray diffraction (XRD, FOCUS, D8 Advance, Bruker, Germany) at a scanning rate of $2°\ min^{-1}$ and a scanning range of 1–10°. The pore structures of the pristine MCM-41 and the modified MCM-41 were determined using Brunner-Emmet-Teller (BET, TriStar3000, Micromeritics Instruments Corporation). The thermostability of the membranes were investigated using a thermogravimetric (TG) analyser (STA409PC, Netzsch Group) in a nitrogen atmosphere at a ramp rate of $10°C\ min^{-1}$. A scanning electron microscope (SEM, S-4800, Hitachi Ltd) was used to observe the samples of the ionic liquid-modified molecular sieves and the surfaces and cross sections of the mixed matrix membranes. Transmission electron microscopy (TEM, JEOL, JEM-2100EX, Japan) was used to characterize the mesoporous structure of MCM-41 and IL-modified MCM-41. Contact angles (CAs) of water on the membranes were measured by using a contact angle analyzer (DSA 100, KRUSS, Germany).

## 2.5. Swelling experiment

The prepared membranes were cut into samples of the same size, which were then dried thoroughly in the drying oven at 60°C. Subsequently, the samples were weighed and immersed into the feeding liquid with a certain concentration at a constant temperature. The membranes were picked up after a set time interval. The remaining liquid on the membrane surfaces was wiped quickly using Kimwipes (Kimberly Clark). Subsequently, the membranes were weighed and placed in the feeding liquid again. This operation was repeated until there was no significant change in the mass of the membrane samples, which suggested that the swelling had reached equilibrium. The swelling degree (DS%) can be calculated by equation (2.1) as follows:

$$DS(\%) = \frac{W_s - W_d}{W_d}, \tag{2.1}$$

where $W_d$ and $W_s$ are the mass (g) of the dry membranes and the mass of samples at swelling equilibrium, respectively.

## 2.6. Pervaporation experiment

In this study, the equipment for the PV experiment was provided by Tianjin University Beiyang Chemical Equipment Co., Ltd. When the feeding liquid was heated to a preset temperature, it was pumped from the liquid feeding tank using a circulating pump. The liquid flowed into the membrane chamber through the flow meter. In the membrane chamber, the feeding liquid was separated using PV. The permeated components, also known as the penetrants, were collected using a liquid nitrogen trap, whereas the remaining liquid was returned to the feeding tank via liquid circulation. The experimental temperature error was controlled to within ±0.5°C. The feeding liquid flow was measured using a rotor flow meter. The vacuum in the downstream side of the membrane chamber was greater than 0.1 MPa; the effective area of the PV membrane in the membrane chamber was $3.6 \times 10^{-3}$ m². A gas chromatograph (GC, 7900, TECHCOMP (HOLDINGS) LIMITED) was used to determine the compositions of the feeding liquid and the penetrants. The chromatographic column was an HP-FFAP (50 m × 0.2 mm × 0.3 µm) capillary column. A hydrogen flame ionization detector was used as the detector. Nitrogen was the carrier gas. The column temperature was 120°C; the temperatures of the sample injector and the detector were 160°C. The separating performance of the PV membrane was primarily evaluated using the separation factor $\alpha$, the permeation flux J (g m$^{-2}$ h$^{-1}$) and the pervaporation separation index (PSI) as follows:

$$\alpha = \frac{(C_i/C_j)_{Permeate}}{(C_i/C_j)_{Feed}}, \tag{2.2}$$

$$J = \frac{W}{At} \tag{2.3}$$

and
$$PSI = J \times (\alpha - 1), \tag{2.4}$$

where $C_i$ and $C_j$ are the mass fractions of butanol and water, respectively, in the penetrant; and $W$, $A$, $t$ and $J$ represent the mass of penetrant (g), the effective area of the membrane (m²), the effective operating time of the PV experiment (h), and the permeation flux (g m$^{-2}$ h$^{-1}$), respectively. PSI is a usually used index to compare the comprehensive separation performance of different membranes.

# 3. Results and discussion

## 3.1. Characterization of the IL-modified MCM-41

### 3.1.1. X-ray photoelectron spectroscopy

The element analysis results before and after the MCM-41 was modified by IL are shown in table 1. The data show that MCM-41 primarily comprises Si and O in mole fractions of 26.5% and 64.1%, respectively. The existence of the element C might be some remaining template agent used in the synthesis of MCM-41. Compared with the pristine MCM-41, the contents of Si and O in the IL-modified MCM-41 decreased, while that of C substantially increased. In addition, IL-specific elements N, F and S were also observed. In particular, the contents of S and F in MCM-41-IL1 were 0.9% and 6.0%, respectively.

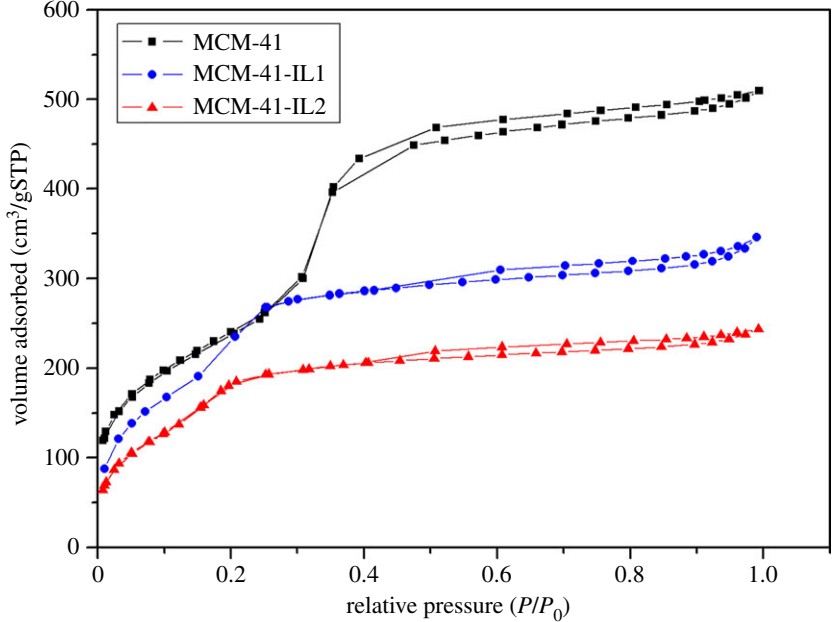

**Figure 2.** $N_2$ adsorption–desorption isotherms of MCM-41 and ionic liquid-modified MCM-41.

**Table 1.** Elemental composition of the molecular sieve before and after modification with ionic liquids by XPS.

| sample | atom percentage (mol%) | | | | | |
|---|---|---|---|---|---|---|
| | C | N | O | Si | S | F |
| MCM-41 | 9.4 | — | 64.1 | 26.5 | — | — |
| MCM-41-IL1 | 34.9 | 3.8 | 42.6 | 11.8 | 0.9 | 6.0 |
| MCM-41-IL2 | 23.0 | 2.3 | 55.3 | 16.3 | 1.1 | 2.0 |

**Table 2.** Structure parameters for molecular sieves and ionic liquid-modified molecular sieves.

| sample | pore diameter (nm) | specific surface area ($m^2 g^{-1}$) | pore volume ($cm^3 g^{-1}$) |
|---|---|---|---|
| MCM-41 | 2.9 | 890 | 0.80 |
| MCM-41-IL1 | 2.4 | 801 | 0.55 |
| MCM-41-IL2 | 2.3 | 780 | 0.38 |

The contents of the IL-specific elements S and F in MCM-41-IL2 were 1.1% and 2.0%. Based on the XPS results, it was assumed that the ionic liquids could be effectively coupled with MCM-41 through hydrogen bonds and the nucleophilic reaction between anions of the ionic liquids and silanized MCM-41.

### 3.1.2. Pore structure analysis

The $N_2$ adsorption–desorption isotherms of MCM-41 and ionic liquid-modified MCM-41 are shown in figure 2. The structural parameters of an IL-modified mesoporous molecular sieve are shown in table 2. Figure 2 and table 2 suggest that the pore diameter, the specific surface area and the pore volume decreased after IL modification. This suggests that some pores were occupied after the IL and MCM-41 were coupled. Because IL2 has a longer side chain (octyl) in the cation than IL1 (ethyl + vinyl), the pore volume of MCM-41-IL2 decreased more than that of MCM-41-IL1. However, the pore diameters of the modified MCM-41 still remained between 2 and 5 nm, which suggests that the mesoporous structure was not damaged.

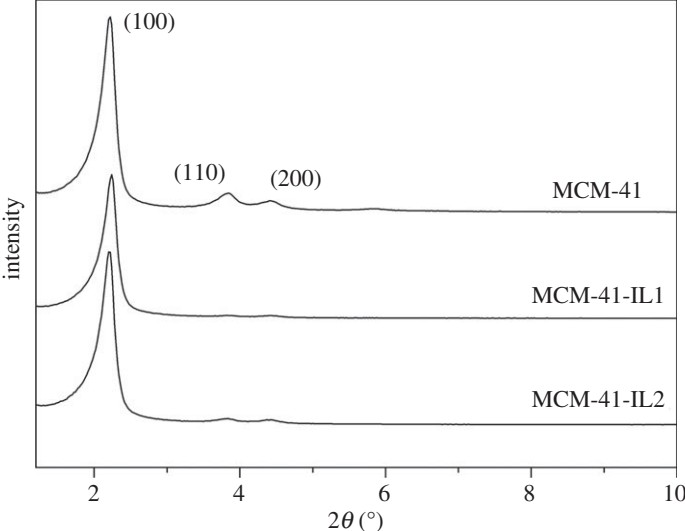

**Figure 3.** XRD patterns of MCM-41 and MCM-41-IL.

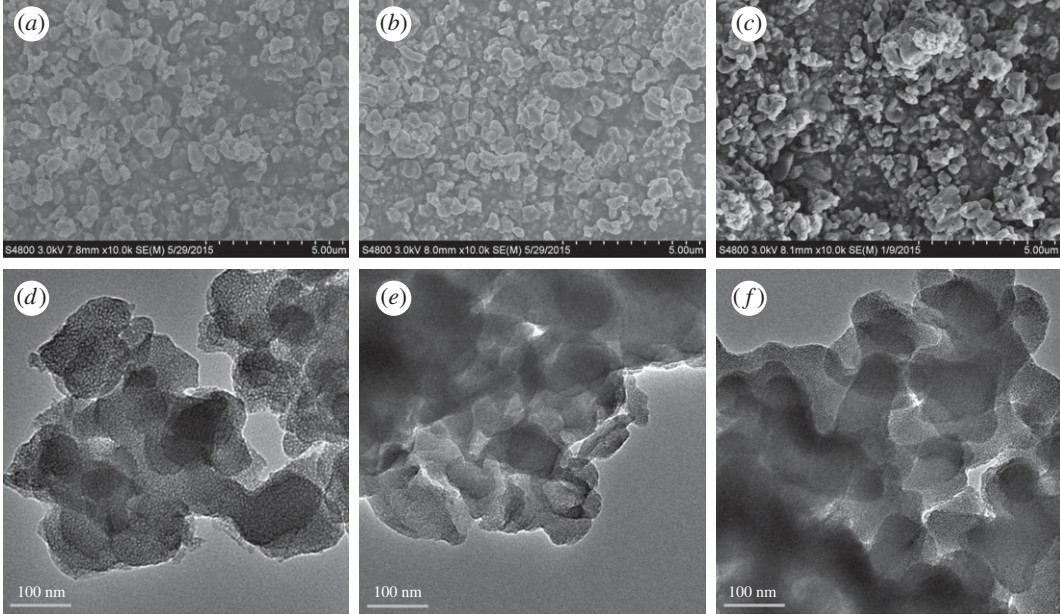

**Figure 4.** SEM images of MCM-41 and ionic liquid-modified MCM-41: (*a*) MCM-41, (*b*) MCM-41-IL1, (*c*) MCM-41-IL2 and TEM images of MCM-41 and ionic liquid-modified MCM-41: (*d*) MCM-41, (*e*) MCM-41-IL1 and (*f*) MCM-41-IL2.

### 3.1.3. X-ray diffraction patterns

The XRD patterns of MCM-41 and IL-modified MCM-41 are shown in figure 3. The data in the figure show that there were marked characteristic peaks of three crystal faces (100, 110 and 200) on MCM-41. In addition, strong diffraction peaks of crystal face (100) were observed on both MCM-41-IL1 and MCM-41-IL2; their diffraction angles remained unchanged. This indicates that the basic structure of MCM-41 was not damaged after modification. However, the characteristic peaks of the IL-modified MCM-41 decreased slightly. One possible reason was that organic groups were attached to the framework of the molecular sieve from the coupling with IL groups (figure 1).

### 3.1.4. Scanning electron microscope

The SEM images of MCM-41 before and after IL modification are shown in figure 4*a*–*c*. The data in the figure show that IL modification made no visible difference to the particle morphology. Crystals were

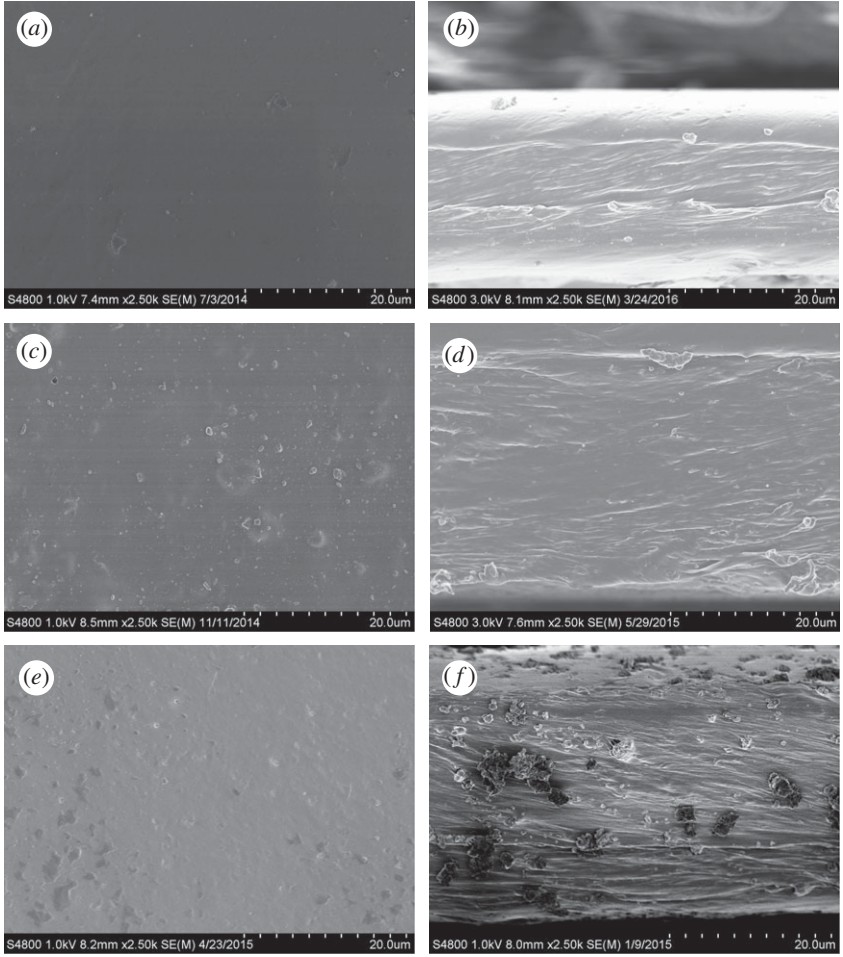

**Figure 5.** SEM images of PEBA and mixed matrix membranes: (*a*) surface of PEBA, (*b*) cross section of PEBA, (*c*) surface of 5% MCM-41-IL1-PEBA, (*d*) cross section of 5% MCM-41-IL1-PEBA, (*e*) surface of 5% MCM-41-IL2-PEBA and (*f*) cross section of 5% MCM-41-IL2-PEBA.

observed before and after the modification. Particle size was also unchanged. The TEM images of MCM-41 before and after IL modification are shown in figure 4*d*–*f*. The mesoporous structure of MCM-41 and modified MCM-41 can be seen. After modification with ionic liquid, the pore size of MCM-41-IL1 and MCM-41-IL2 decreased a little.

## 3.2. Characterization of the membranes

### 3.2.1. Scanning electron microscope

The SEM images of the surfaces and cross sections of the PEBA and the modified PEBA membranes are shown in figure 5. It can be seen from figure 5*a*,*b* that the pristine PEBA membrane was flat at its surface with a uniform cross section. Figures 5*c*,*e* are the SEM images of the surfaces of modified PEBA membranes with different types of IL. The surface of 5% MCM-41-IL2-PEBA was smoother than that of 5% MCM-41-IL1-PEBA. As shown in figure 5*d*,*f*, the molecular sieves were tightly wrapped by PEBA and evenly distributed on the membrane surface or cross section; the membranes were dense without any defects.

### 3.2.2. Swelling experiment

The variation of the degree of swelling for the modified PEBA membrane in the butanol solution with a mass fraction of 2.5 wt% over time is shown in figure 6. For the first 2 h, the degrees of swelling for both

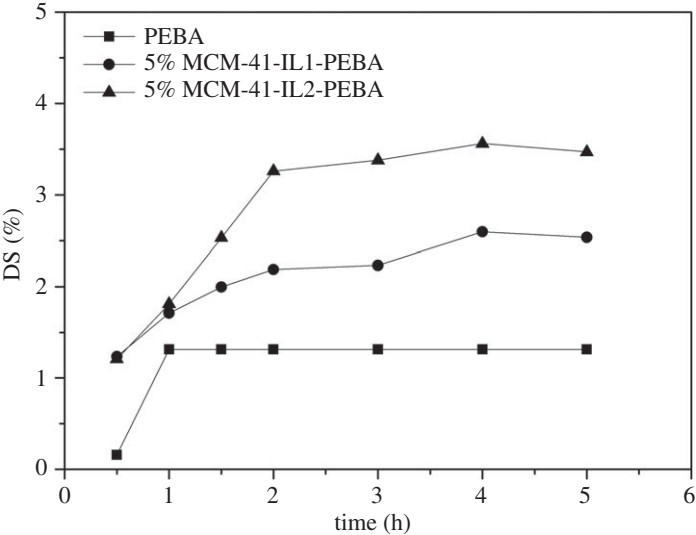

**Figure 6.** Degree of swelling for different membranes (2.5 wt% butanol, 35°C).

modified membranes continued to increase with time, basically reaching swelling equilibrium in 2 h. The degree of swelling for the 5% MCM-41-IL2-PEBA was higher than that for the MCM-41-IL1-PEBA membrane because [OMPY][Tf$_2$N] was more selective to butanol [33].

The degrees of swelling for mixed matrix membranes with different filling amounts of MCM-41-IL1 and MCM-41-IL2 in pure water and butanol solution (2.5 wt%) at the temperature of 35°C are shown in figure 7a,b. The data show that the degrees of swelling for the modified membranes increased in both butanol solution and pure water as the filling amounts of the two modified MCM-41 samples increased. With the increase of filling amount, water molecules tended to form hydrogen bonding with the Si–OH on the molecular sieve [34]. Therefore, the degree of swelling for the modified membrane in pure water increased with the filling amount. Comparing the degrees of swelling for the same mixed matrix membrane in pure water and in a butanol solution suggests that the degrees of swelling in the butanol solution were higher than those in pure water. The reason is that butanol is preferably adsorbed by PEBA [33]. In addition, organophilic ionic liquid enhanced the sorption of butanol on mixed matrix membranes.

### 3.2.3. Thermogravimetric analysis

The thermogravimetric (TG) analyses of PEBA, 5% MCM-41-PEBA, 5% MCM-41-IL1-PEBA, and 5% MCM-41-IL2-PEBA are shown in figure 8, which indicate that the PEBA membrane was highly thermostable because it started decomposing at approximately 210°C. The thermal decomposition temperature of the membrane with IL-modified MCM-41 was near that of the PEBA membrane. This suggested that the addition of IL-modified MCM-41 made little difference to the thermostability of the PEBA membrane. Because 210°C is much greater than the PV process temperature, the modified membrane prepared in this study could meet the pervaporation requirements.

### 3.2.4. Water contact angle measurement

Figure 9a–d shows the water CAs for the pristine PEBA membrane and the mixed matrix membranes. The water CAs for the pristine PEBA membrane was 77°. Since molecular sieve MCM-41 has abundant –OH, the water CA of the 5% MCM-41-PEBA membrane decreased to 60°, which meant that the hydrophilicity of 5% MCM-41-PEBA membrane was increased. The anion of IL1 and IL2 shows strong hydrophobicity because it has two trifluoromethyl (-CF3) groups. With the incorporation of ionic liquid-modified MCM-41, the water CAs of 5% MCM-41-IL1-PEBA membrane and 5% MCM-41-IL2-PEBA membrane increased to 85° and 88°, respectively, which meant that the membranes' hydrophobicity was enhanced. Due to IL2 has longer alkyl chain length in its cation, the hydrophobicity of 5% MCM-41-IL2-PEBA membrane was higher than that of 5% MCM-41-IL1-PEBA membrane.

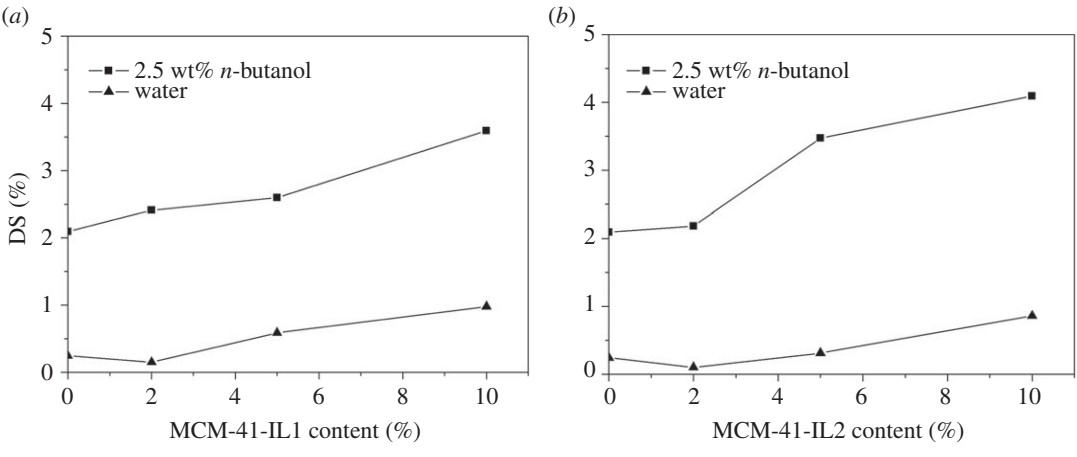

**Figure 7.** Effect of the IL-modified MCM-41 content on DS (2.5 wt% butanol, 35°C): (*a*) MCM-41-IL1 and (*b*) MCM-41-IL2.

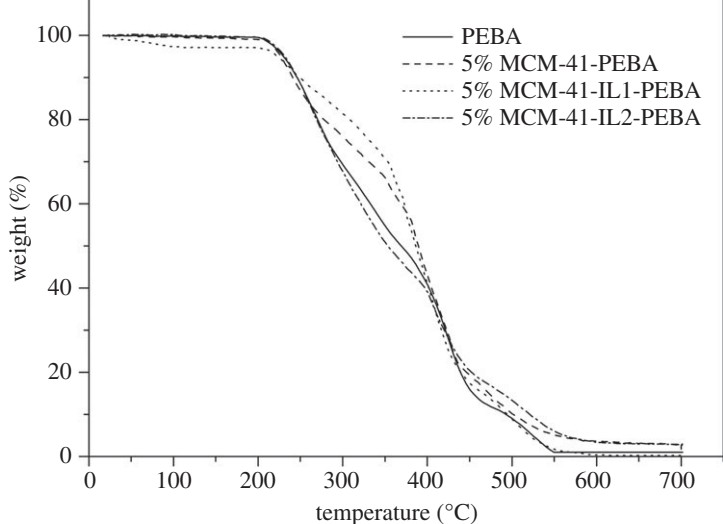

**Figure 8.** TG analysis curves of MCM-41-IL-PEBA membranes.

## 3.3. Pervaporation experiment

### 3.3.1. Effect of IL-modified MCM-41 loading

The influence of the MCM-41-IL1 contents on the PV performance is shown in figure 10*a*, where the temperature and mass fraction of the feeding liquid are 35°C and 2.5 wt%, respectively. Compared with the pristine PEBA membrane, the permeation flux and separation factor of the modified PEBA membrane were both enhanced when there was an appropriate amount of IL-modified MCM-41. On the one hand, the mesoporous structure of the molecular sieve facilitated butanol and water diffusion in the membrane, for which the permeation flux of the modified membrane was significantly improved. On the other hand, the IL-modified MCM-41 showed favourable butanol selectivity, which would promote butanol sorption. Therefore, the separation factor of the mixed matrix membrane was also enhanced. When the filling amount of MCM-41-IL1 was 5%, the modified PEBA membrane showed the best PV separation performance; its separation factor and permeation flux were 22.7 and 410 g m$^{-2}$ h$^{-1}$, respectively. However, the data in figure 10*a* show that the permeation flux and separation factor of the mixed matrix membrane decreased as the content of MCM-41-IL1 continued to increase. Clustering occurred when the filling amount of MCM-41-IL1 in the modified membrane was 10%. The molecular sieve was not uniformly dispersed in the membrane and the transfer path of the molecule was extended.

The incorporation of MCM-41-IL2 had a similar effect on pervaporation performance. The influence of the filling amount of MCM-41-IL2 on the PV separation performance is shown in figure 10*b*. The data show that the permeation flux and separation factor of the MCM-41-IL2-modified PEBA membrane were both

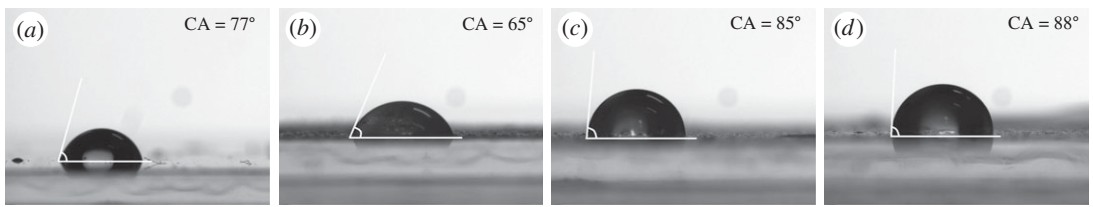

**Figure 9.** Water CA of different membranes (*a*) Pristine PEBA membrane, (*b*) 5% MCM-41-PEBA membrane, (*c*) 5% MCM-41-IL1-PEBA membrane and (*d*) 5% MCM-41-IL2-PEBA membrane.

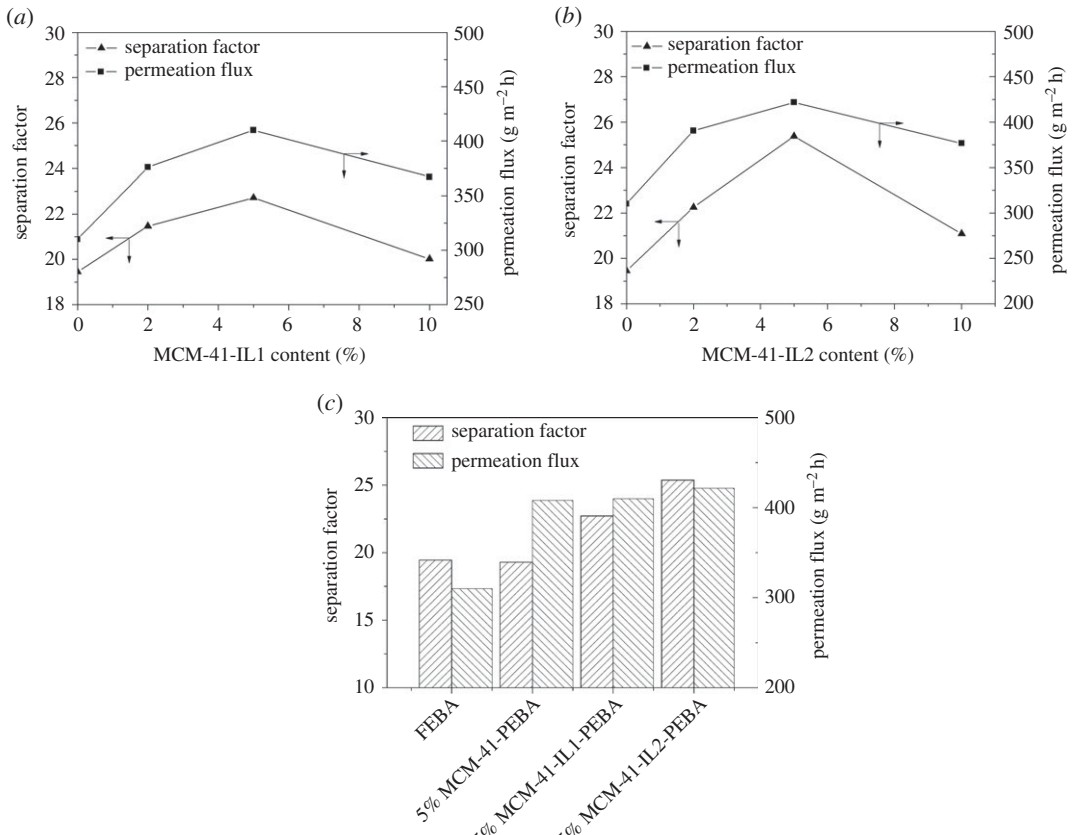

**Figure 10.** (*a*) Effect of MCM-41-IL1 content on the pervaporation performance of MCM-41-IL1-PEBA membranes; (*b*) effect of MCM-41-IL2 content on the pervaporation performance of MCM-41-IL2-PEBA membranes and (*c*) pervaporation performance comparison of different membranes.

significantly greater than those of the pristine PEBA membrane. When the filling content was 5 wt%, the permeation flux and separation factor of the modified PEBA membrane peaked at 421.7 g m$^{-2}$ h$^{-1}$ and 25.4, respectively. The IL-modified molecular sieve played an important role in enhancing the capacity of the membrane to selectively separate butanol and promote the adsorption and diffusion of small molecules in the membrane.

The PV separation performances of pristine PEBA, 5% MCM-41-PEBA, 5% MCM-41-IL1-PEBA and 5% MCM-41-IL2-PEBA are compared in figure 10*c*. According to the pore structure data of the molecular sieves shown in table 2, MCM-41 and IL-modified MCM-41 have mesoporous structures. The free volume of the PEBA membrane could be enlarged when MCM-41 or IL-modified MCM-41 was added as the filling agent. Therefore, the permeation flux of the mixed matrix membrane could be enhanced.

Due to the increased hydrophilicity of 5% MCM-41-PEBA membrane, its separation factor of butanol/water was a little lower than that of pristine PEBA membrane. According to the research for butanol extraction by Ha *et al.* [26], the butanol/water selectivity strongly depends on the hydrophobicity of anions of ILs followed by the hydrophobicity of cations of ILs. [Tf2N]$^-$-based ionic liquids showed best

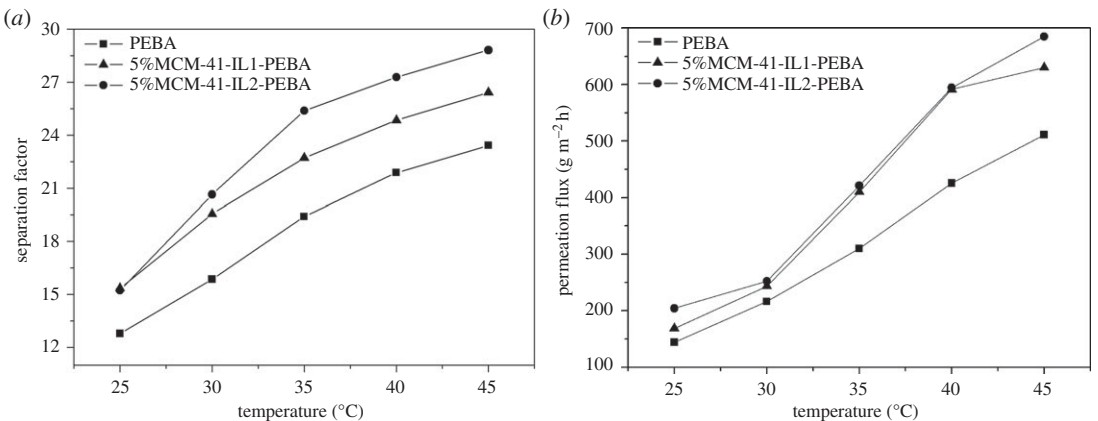

**Figure 11.** (*a*) Effect of the liquid temperature on the separation factor of PEBA-modified membranes and (*b*) effect of the liquid temperature on the permeation flux of PEBA-modified membranes.

extraction efficiency for butanol recovery in their research. And the hydrophobicity of cations would increase as the cation alkyl chain length increases. Both IL1 and IL2 are [Tf2N]⁻-based ionic liquids with high hydrophobicity, therefore the separation factor of butanol/water could be enhanced when IL-modified MCM-41 was used to fill the membrane. IL2 performed better in selective butanol pervaporation because it has longer alkyl chain length in its cation. The pervaporation performance of these membranes was also consistent with the swelling experiment and water CA measurement results.

### 3.3.2. Effect of feed temperature

The influence of feed temperature on the separation factors and permeation fluxes of the pristine PEBA membrane, 5% MCM-41-IL1-PEBA and 5% MCM-41-IL2-PEBA is shown in figure 11*a*,*b*. The data show that the separation factors and permeation fluxes of both the pristine PEBA membrane and the mixed matrix membrane increased with feed temperature. One reason for this behaviour was that the small molecules in the feed moved faster and diffused in the membrane at a higher rate as the feed temperature increased. Another reason was that the segment movement of the PEBA matrix accelerated and the free volume of the membrane increased with temperature, which promoted the mass transfer process.

### 3.3.3. Stability of MCM-41-IL-PEBA membrane

According to a previous study by our research group, the separation factor of the IL-blended PEBA membrane decreased markedly with IL loss after it was used for 26 h [33]. However, the variations of the permeation flux and separation factor of 5% MCM-41-IL2-PEBA with time were within a small range (figure 12). Within 100 h of operation, the permeation flux and the separation factor of 5% MCM-41-IL2-PEBA were 25.2 and 414.1 g m$^{-2}$ h$^{-1}$ on average, both of which were higher than those of the pristine PEBA membrane and showed favourable stability. This suggests that IL modification of MCM-41 is also an effective way to immobilize ionic liquid.

### 3.3.4. Comparison with other pervaporation membranes

A comparison of separation performance of the MCM-41-IL-PEBA membranes in our work and some other membranes reported in the literature is shown in table 3. In order to evaluate the comprehensive performance of membranes, PSI values are calculated and listed. Under the similar experimental conditions, MCM-41-IL-PEBA membranes showed good pervaporation performance (table 3). PSI of 10 289 g m$^{-2}$ h$^{-1}$ was obtained by using the 5% MCM-41-IL2-PEBA membrane, which was quite high compared with other membranes. In the view of the overall techno economic feasibility of pervaporation, the 5% MCM-41-IL2-PEBA membrane would be a promising practical membrane for butanol separation or recovery.

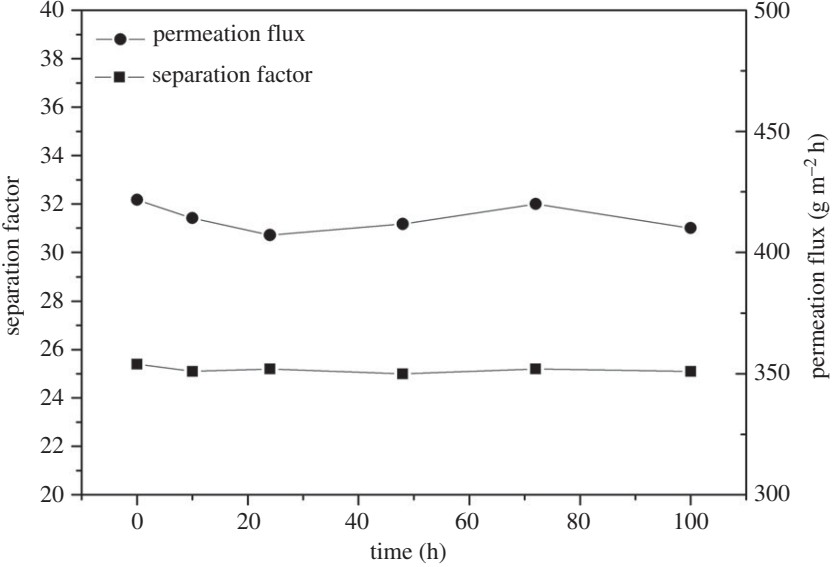

**Figure 12.** Effect of operating time on pervaporation performance of 5% MCM-41-IL2-PEBA membranes.

**Table 3.** Comparison of separation performance with some reported membranes on butanol pervaporation.

| membrane | feed butanol content (wt%) | $T$ (°C) | $J$ (g m$^{-2}$ h$^{-1}$) | $\alpha$ | PSI (g m$^{-2}$ h$^{-1}$) | ref. |
|---|---|---|---|---|---|---|
| CNT-PEBA | 0.8% | 37 | 153 | 19.4 | 2815 | [18] |
| ZIF-71-PEBA | 1 | 37 | 520 | 18.8 | 9256 | [21] |
| PTFE | 1.25 | 40 | 170 | 8.5 | 1275 | [35] |
| HTPB-PU | 3 | 40 | 19.0 | 21.3 | 386 | [36] |
| PDMS (Pervatech) | 5 | 25 | 851.9 | 9 | 6815 | [37] |
| PDMS-PVDF | 1.5 | 37 | 158.2 | 17.3 | 2579 | [38] |
| PTMSP | 1 | 25 | 60 | 52 | 3060 | [39] |
| PDMS/PE/Brass | 2 | 37 | 132 | 32 | 4092 | [40] |
| c-PDMS/BPPO | 5 | 40 | 220 | 35 | 7480 | [41] |
| PDMS/ZSM-5 | 1.5 | 37 | 99.8 | 16.7 | 1567 | [42] |
| 5% MCM-41-IL1-PEBA | 2.5 | 35 | 410 | 22.7 | 8897 | this work |
| 5% MCM-41-IL2-PEBA | 2.5 | 35 | 421.7 | 25.4 | 10 289 | this work |

# 4. Conclusion

Hydrophobic ionic liquids 1-ethyl-3-vinylimidazolium bis[(trifluoromethyl)sulfonyl]imide (IL1) and N-octyl-pyridinium bis((trifluoromethyl)sulfonyl)imide (IL2) were used to modify the mesoporous molecular sieve MCM-41. XPS characterization results indicated that IL-specific elements S and F were observed in IL-modified molecular sieves. According to XRD and BET analyses, the porous structure of the molecular sieve remained unchanged after IL was introduced while the specific surface area, pore volume and pore diameter of the modified MCM-41 decreased slightly.

IL-modified MCM-41 was used as the filling agent to prepare MCM-41-IL-PEBA mixed matrix membranes with different filling contents. The incorporation of MCM-41-IL made little difference to membrane thermostability, i.e. the membrane was able to meet the requirements for PV separation. Butanol was preferentially adsorbed by the MCM-41-IL-PEBA mixed matrix membranes. Compared with the pristine PEBA membrane, the MCM-41-IL-PEBA mixed matrix membranes showed enhanced performance for butanol pervaporation when the filling content was appropriate. In all the membranes investigated in this research, MCM-41-IL2-PEBA with a filling content of 5% showed the

highest permeation flux of 421.7 g m$^{-2}$ h$^{-1}$ and separation factor of 25.4. The PV separation performance of MCM-41-IL-PEBA increased with feed temperature. The 5% MCM-41-IL2-PEBA performed stably within 100 h of operation, with its separation factor and permeation flux reaching 25.2 and 414.1 g m$^{-2}$ h$^{-1}$ on average.

Data accessibility. The datasets supporting this article have been uploaded as part of the electronic supplementary material.

Authors' contributions. Y.F.L. contributed to the conception and drafting of the article; D.D.Y. contributed to the acquisition of data and analysis of data; Y.H.W. guided the project and revised the final manuscript and she is the corresponding author of this article. All authors gave final approval for publication.

Competing interests. We declare we have no competing interests.

Funding. Financial support comes from the National Natural Science Foundation of China (NSFC no. 21446002) and the Natural Science Foundation of Shanghai (no. 16ZR1438300). (Both hosted by Y.H.W.)

Acknowledgements. The authors thank the sponsorship of National Natural Science Foundation of China, Shanghai Natural Science Foundation.

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
