## [Reviewer comments · Royal Society Open Science]

Review History

RSOS-190291.R0 (Original submission)

Review form: Reviewer 1

Is the manuscript scientifically sound in its present form?

Yes

Are the interpretations and conclusions justified by the results?

Yes

Is the language acceptable?

Yes

Is it clear how to access all supporting data?

Yes

Do you have any ethical concerns with this paper?

No

Have you any concerns about statistical analyses in this paper?

No

Recommendation?

Major revision is needed (please make suggestions in comments)

Comments to the Author(s)

In this submitted manuscript, the authors described the preparation of ionic liquid-modified MCM-41-polymer mixed matrix membrane for butanol pervaporation. They characterized the formed membranes in detail and studied their performance on butanol pervaporation. Some useful results were contained in this manuscript. In my opinion, this manuscript could be accepted after some modifications.

1. The contents of the two ionic liquids in MCM-41 should be determined by elemental analysis because the XPS method was inaccurate.
2. The authors said that “the ionic liquids could be effectively coupled with MCM-41 through hydrogen bonds and the nucleophilic reaction between anions of the ionic liquid and silanized MCM-41.” Was there any evidence for the the nucleophilic reaction between anions of the ionic liquid and silanized MCM-41? Or how the reaction occurred?
3. TEM of the prepared IL-modified MCM-41 should be conducted. Meanwhile, the used membrane should also be characterized.
4. The results of the prepared membrane for butanol pervaporation should be compared with some reported results of the similar materials.
5. Some reasonable explanations should be provided for the different performance of the two ILs modified materials.
6. The results of the non-modified MCM-41 for butanol pervaporation should be provided.

Review form: Reviewer 2 (Xiang Zhu)

Is the manuscript scientifically sound in its present form?

Yes

Are the interpretations and conclusions justified by the results?

Yes

Is the language acceptable?

Yes

Is it clear how to access all supporting data?

Yes

Do you have any ethical concerns with this paper?

No

Have you any concerns about statistical analyses in this paper?

No

Recommendation?

Accept with minor revision (please list in comments)

Comments to the Author(s)

This is an interesting study that reports the used of ILs as effective fillers for the fabrication of mixed matrix membranes. The as-obtained membranes exhibit interesting butanol pervaporation

performance. Membrane-based separations are becoming more and more important and attractive. The materials including MCM-hybrids and membranes reported in this manuscript are well characterized and reasonable membrane separation experiments are performed. Overall, this manuscript is well organized and could be published after revisions.

- 1) BET characterization and pore size distribution should be provided along with N₂-adsorption curves. The authors should exhibit these adsorption-desorption curves, not just list data in a table.
- 2) What is the purity of these two ionic liquids used in this manuscript? Please provide NMR data. As the authors comment that they are hydrophobic. Please show their contact angle experiments and results. Furthermore, do ILs contain any water? Will water affect the membrane-based separation performance?
- 3) It seems that the authors just load each IL with a certain amount. Did the authors systematically study the IL effect? For example, did the author load more ILs into MCM-41 to get more ILs incorporated into the matrix of MCM-41? The effect of ILs on the butanol pervaporation should be explored.
- 4) What is the mechanism for the enhanced membrane-based separation performance? The authors should perform more experiments and get more better understanding.

Decision letter (RSOS-190291.R0)

05-Apr-2019

Dear Dr Wu:

Title: Ionic Liquid Modified MCM-41-Polymer Mixed Matrix Membrane for Butanol Pervaporation
Manuscript ID: RSOS-190291

The editor assigned to your manuscript has now received comments from reviewers. We would like you to revise your paper in accordance with the referee and Subject Editor suggestions which can be found below (not including confidential reports to the Editor). Please note this decision does not guarantee eventual acceptance.

Please submit your revised paper before 28-Apr-2019. Please note that the revision deadline will expire at 00.00am on this date. If we do not hear from you within this time then it will be assumed that the paper has been withdrawn. In exceptional circumstances, extensions may be possible if agreed with the Editorial Office in advance. We do not allow multiple rounds of revision so we urge you to make every effort to fully address all of the comments at this stage. If deemed necessary by the Editors, your manuscript will be sent back to one or more of the original reviewers for assessment. If the original reviewers are not available we may invite new reviewers.

Please also include the following statements alongside the other end statements. As we cannot publish your manuscript without these end statements included, if you feel that a given heading is not relevant to your paper, please nevertheless include the heading and explicitly state that it is not relevant to your work.

- Ethics statement

Please clarify whether you received ethical approval from a local ethics committee to carry out your study. If so please include details of this, including the name of the committee that gave consent in a Research Ethics section after your main text. Please also clarify whether you received informed consent for the participants to participate in the study and state this in your Research Ethics section.

OR

Please clarify whether you obtained the necessary licences and approvals from your institutional animal ethics committee before conducting your research. Please provide details of these licences and approvals in an Animal Ethics section after your main text.

OR

Please clarify whether you obtained the appropriate permissions and licences to conduct the fieldwork detailed in your study. Please provide details of these in your methods section.

- Acknowledgements

RSC Associate Editor: 1
Comments to the Author:
(There are no comments.)

RSC Associate Editor: 2
Comments to the Author:
(There are no comments.)

Reviewers' Comments to Author:
Reviewer: 1

Comments to the Author(s)

In this submitted manuscript, the authors described the preparation of ionic liquid-modified MCM-41-polymer mixed matrix membrane for butanol pervaporation. They characterized the formed membranes in detail and studied their performance on butanol pervaporation. Some useful results were contained in this manuscript. In my opinion, this manuscript could be accepted after some modifications.

1. The contents of the two ionic liquids in MCM-41 should be determined by elemental analysis because the XPS method was inaccurate.
2. The authors said that "the ionic liquids could be effectively coupled with MCM-41 through hydrogen bonds and the nucleophilic reaction between anions of the ionic liquid and silanized MCM-41." Was there any evidence for the the nucleophilic reaction between anions of the ionic liquid and silanized MCM-41? Or how the reaction occurred?
3. TEM of the prepared IL-modified MCM-41 should be conducted. Meanwhile, the used membrane should also be characterized.
4. The results of the prepared membrane for butanol pervaporation should be compared with some reported results of the similar materials.
5. Some reasonable explanations should be provided for the different performance of the two ILs modified materials.
6. The results of the non-modified MCM-41 for butanol pervaporation should be provided.

Reviewer: 2

Comments to the Author(s)

This is an interesting study that reports the used of ILs as effective fillers for the fabrication of mixed matrix membranes. The as-obtained membranes exhibit interesting butanol pervaporation performance. Membrane-based separations are becoming more and more important and attractive. The materials including MCM-hybrids and membranes reported in this manuscript are well characterized and reasonable membrane separation experiments are performed. Overall, this manuscript is well organized and could be published after revisions.

- 1) BET characterization and pore size distribution should provided along with N₂-adsorption curves. The authors should exhibit these adsorption-desorption curves, not just list data in a table.
- 2) What is the purity of these two ionic liquids used in this manuscript? Please provide NMR data. As the authors comment that they are hydrophobic. Please show their contact angle experiments and results. Furthermore, do ILs contain any water? Will water affect the membrane-based separation performance?
- 3) It seems that the authors just load each IL with a certain amount. Did the authors systematically study the IL effect? For example, did the author load more ILs into MCM-41 to get more ILs incorporated into the matrix of MCM-41? The effect of ILs on the butanol pervaporation should be explored.

4) What is the mechanism for the enhanced membrane-based separation performance? The authors should perform more experiments and get more better understanding.

Author's Response to Decision Letter for (RSOS-190291.R0)

See Appendix A.

RSOS-190291.R1 (Revision)

Review form: Reviewer 1

Is the manuscript scientifically sound in its present form?

Yes

Are the interpretations and conclusions justified by the results?

Yes

Is the language acceptable?

Yes

Is it clear how to access all supporting data?

Yes

Do you have any ethical concerns with this paper?

No

Have you any concerns about statistical analyses in this paper?

No

Recommendation?

Accept as is

Comments to the Author(s)

My confusion has been cleared on the basis of the authors' responses. Thus, I recommended the revised manuscript to be accepted.

Review form: Reviewer 2 (Xiang Zhu)

Is the manuscript scientifically sound in its present form?

Yes

Are the interpretations and conclusions justified by the results?

Yes

Is the language acceptable?

Yes

Do you have any ethical concerns with this paper?

No

Recommendation?

Accept as is

Comments to the Author(s)

All my concerns and the other reviewers' questions have been well addressed by the authors. They have put lots of efforts on this and the quality of this manuscript has been greatly improved. As a result, it can be accepted in its current form now.

Decision letter (RSOS-190291.R1)

01-Jul-2019

Dear Dr Wu:

Title: Ionic Liquid-Modified MCM-41-Polymer Mixed Matrix Membrane for Butanol Pervaporation

Manuscript ID: RSOS-190291.R1

It is a pleasure to accept your manuscript in its current form for publication in Royal Society Open Science. The chemistry content of Royal Society Open Science is published in collaboration with the Royal Society of Chemistry.

Yours sincerely,

Dr Laura Smith

Publishing Editor, Journals

Royal Society of Chemistry

Thomas Graham House

Science Park, Milton Road

Cambridge, CB4 0WF

Royal Society Open Science - Chemistry Editorial Office

RSC Associate Editor:
Comments to the Author:
I apologise this has taken longer than usual.

RSC Subject Editor:
Comments to the Author:
(There are no comments.)

Reviewer(s)' Comments to Author:
Reviewer: 1

Comments to the Author(s)
My confusion has been cleared on the basis of the authors' responses. Thus, I recommended the revised manuscript to be accepted.

Reviewer: 2

Comments to the Author(s)
All my concerns and the other reviewers' questions have been well addressed by the authors. They have put lots of efforts on this and the quality of this manuscript has been greatly improved. As a result, it can be accepted in its current form now.

Appendix A

Response to Referees

Thank you very much for helpful and valuable comments. We supplemented some experiments and made corrections carefully. The required statements are also included in the revised manuscript. Some new figures are added and more references are cited in the revision. Therefore, the sequence number of the figures and the references are changed accordingly. The manuscript was revised according to your specific comments. Detailed explanations for the comments are shown as follows.

Reviewer: 1

Comments to the Author(s)

In this submitted manuscript, the authors described the preparation of ionic liquid-modified MCM-41-polymer mixed matrix membrane for butanol pervaporation. They characterized the formed membranes in detail and studied their performance on butanol pervaporation. Some useful results were contained in this manuscript. In my opinion, this manuscript could be accepted after some modifications.

1. The contents of the two ionic liquids in MCM-41 should be determined by elemental analysis because the XPS method was inaccurate.

Response: Thank you for your suggestion. Ionic liquid modified MCM-41 is a solid sample. It's difficult to measure the contents of ionic liquids in a solid sample accurately. In revision, we also characterized the ionic liquid modified MCM-41 with EDS (energy dispersive spectrometry). The EDS data was listed in Table C1. Although the two sets of data have some differences due to the detecting depth of XPS and EDS are different. The detecting depth of EDS is about several micrometers, which is much higher than the detecting depth of XPS. Both of the elementary composition results indicated that there was fluorine (characteristic element of the ionic liquids IL1 and IL2) in the modified MCM-41, which meant that ionic liquid was incorporated with MCM-41 effectively.

Table C1 Elemental composition of the molecular sieve by EDS before and after modification with ionic liquids

Sample	Atom percentage (mol%)					
	C	N	O	Si	S	F
MCM-41	12.6	-	60.4	27.0	-	-
MCM-41-IL1	30.1	2.1	52.3	13.7	0.1	1.7
MCM-41-IL2	21.6	1.0	56.0	19.9	0.1	1.4

2. The authors said that “the ionic liquids could be effectively coupled with MCM-41 through hydrogen bonds and the nucleophilic reaction between anions of the ionic liquid and silanized MCM-41.” Was there any evidence for the the

nucleophilic reaction between anions of the ionic liquid and silanized MCM-41?
Or how the reaction occurred?

Response: Figure 1 shows the preparation routes of ionic liquid modified MCM-41. MCM-41 was first been silanized by 3-chloropropyltriethoxysilane. Because the anion [Tf₂N]⁻ of the two ionic liquids is a good nucleophile. According to reference [31, 32], the anion [Tf₂N]⁻ of the two ionic liquids used in our research is a good nucleophile, it can react with the silanized MCM-41 (nucleophilic substitution). On the other hand, ionic liquids can also be coupled with MCM-41 by hydrogen bond. XPS characterization verified that ionic liquid coupled with MCM-41 effectively.

Figure 1. Preparation scheme of ionic liquid modified MCM-41 and the structure of IL1 and IL2

Reference:

31. Hendrickson, J. B., Bair, K. W., Bergeron, R., Giga, A., Skipper, P. L., Sternbach, D. D., Wareing, J. A. 1977 Uses of the triflyl group in organic synthesis. *A review. Org. Prep. Proced. Int.* **9**, 173–207. (doi:10.1080/00304947709356878)
32. Shainyan, B. A., Tolstikova, L. L. 2012 Trifluoromethanesulfonamides and related compounds. *Chem. Rev.* **113**, 699–733. (doi:10.1021/cr300220h)

3. TEM of the prepared IL-modified MCM-41 should be conducted. Meanwhile, the used membrane should also be characterized.

Response: Thank you for your suggestion. We have characterized MCM-41 and ionic liquid modified MCM-41 with TEM. And the TEM images are listed in Figure 2 d), e), f). The mesoporous structure of MCM-41 and ionic liquid modified MCM-41 can be seen

in the TEM images. And after modification, the pore size decreased slightly.

Figure 2. TEM images of MCM-41 and ionic liquid-modified MCM-41: d) MCM-41; e) MCM-41-IL1; f) MCM-41-IL2

In membrane field, the stability of the membrane is usually tested by longtime process. In this research, we conducted a 100 h pervaporation experiment with the mixed matrix membrane. Figure 12 indicated MCM-41-IL-PEBA membrane was of good stability. We characterized the used MCM-41-IL PEBA membranes (after 100 h of pervaporation) by SEM (Figure C1). The surface of the used ionic liquid modified membrane changed little. The IL modified MCM-41 is still incorporated with the polymer PEBA effectively.

a)

b)

Figure C1 SEM images of used membranes a) 5%MCM-41-IL1-PEBA membrane b) 5%MCM-41-IL2-PEBA membrane

4. The results of the prepared membrane for butanol pervaporation should be compared with some reported results of the similar materials.

Response: Thank you for your suggestion. The results of the prepared membrane for butanol pervaporation have been compared with some reported results of other membranes (Table 3). Under the similar experimental conditions, MCM-41-IL-PEBA membranes showed good pervaporation performance. Pervaporation separation index (*PSI*) of 10289 g/m²·h was obtained by using the 5%MCM-41-IL2-PEBA membrane, which was quite high compared with other membranes.

Table 3. Comparison of separation performance with some reported membranes on butanol pervaporation

Membrane	Feed butanol content (wt%)	T(°C)	J (g/m ² ·h)	α	PSI (g/m ² ·h)	Ref
CNT-PEBA	0.8%	37	153	19.4	2815	[18]
ZIF-71-PEBA	1	37	520	18.8	9256	[21]
PTFE	1.25	40	170	8.5	1275	[35]
HTPB-PU	3	40	19.0	21.3	386	[36]
PDMS(Pervatech)	5	25	851.9	9	6815	[37]
PDMS-PVDF	1.5	37	158.2	17.3	2579	[38]
PTMSP	1	25	60	52	3060	[39]
PDMS/PE/Brass	2	37	132	32	4092	[40]
c-PDMS/BPPO	5	40	220	35	7480	[41]
PDMS/ZSM-5	1.5	37	99.8	16.7	1567	[42]
5%-MCM-41-IL1-PEBA	2.5	35	410	22.7	8897	This work
5%-MCM-41-IL2-PEBA	2.5	35	421.7	25.4	10289	This work

5. Some reasonable explanations should be provided for the different performance of the two ILs modified materials.

Response: Thank you for your suggestion. According to the research for butanol extraction by Ha et al [26], the butanol/water selectivity strongly depends on the hydrophobicity of anions of ILs followed by the hydrophobicity of cations of ILs. [Tf2N]⁻ based ionic liquids showed best extraction efficiency for butanol recovery in their research. And the hydrophobicity of cations would increase as the cation alkyl chain length increases. Both IL1 and IL2 are [Tf2N]⁻ based ionic liquids with high hydrophobicity, therefore the separation factor of butanol/water could be enhanced when IL-modified MCM-41 was used to fill the membrane. IL2 performed better in

selective butanol pervaporation because it has longer alkyl chain length in its cation. The pervaporation performance of these membranes was also consistent with the swelling experiment and water contact angle measurement results.

Reference

26. Ha, S. H., Mai, N. L., Koo, Y.-M. 2010 Butanol recovery from aqueous solution into ionic liquids by liquid–liquid extraction. *Process Biochem.* **45**, 1899–1903. (doi:10.1016/j.procbio.2010.03.030)

6. The results of the non-modified MCM-41 for butanol pervaporation should be provided.

Response: Thank you for your suggestion. The result of the non-modified MCM-41 for butanol pervaporation was compared with MCM-41-IL1-PEBA and MCM-41-IL2-PEBA membrane in Figure 10 c). Since MCM-41 has abundant hydrophilic –OH, the hydrophilicity of 5%MCM-41-PEBA membrane was increased. Therefore its separation factor of butanol/water was a little lower than that of the pristine PEBA membrane. On the other hand, the mesoporous structure of MCM-41 facilitated butanol and water diffusion in the membrane, for which the permeation flux of the 5%MCM-41-PEBA membrane was significantly improved. The incorporation of IL-modified MCM-41 is beneficial to the enhancement of separation factor and permeation flux.

Figure 10 c) Pervaporation performance comparison of different membranes

Reviewer: 2

Comments to the Author(s)

This is an interesting study that reports the used of ILs as effective fillers for the fabrication of mixed matrix membranes. The as-obtained membranes exhibit interesting butanol pervaporation performance. Membrane-based separations are becoming more and more important and attractive. The materials including MCM-hybrids and membranes reported in this manuscript are well characterized and reasonable membrane separation experiments are performed. Overall, this manuscript is well organized and could be published after revisions.

1) BET characterization and pore size distribution should be provided along with N₂-adsorption curves. The authors should exhibit these adsorption-desorption curves, not just list data in a table.

Response: Thank you for your suggestion. N₂ adsorption-desorption curves are listed in Figure 2. Figure 2 and Table 2 showed that after IL modification, MCM-41-IL1 and MCM-41-IL2 can keep the mesoporous structure. The pore volume and pore size of the IL-modified MCM-41 decreased a little.

Figure 2. N₂ adsorption-desorption isotherms of MCM-41 and ionic liquid modified MCM-41

2) What is the purity of these two ionic liquids used in this manuscript? Please provide NMR data. As the authors comment that they are hydrophobic. Please show their contact angle experiments and results. Furthermore, do ILs contain any water? Will water affect the membrane-based separation performance?

Response: The ionic liquids were produced by the Shanghai Cheng Jie Chemical Co. LTD. The purity of these two ionic liquids used in this manuscript is higher than 99%. The water content in these two ionic liquids is very low (< 1000 ppm). We have characterized the ionic liquids with NMR. The NMR data of these two ionic liquids are listed as the following:

IL1, Ionic liquid [EVIM][Tf2N]

¹H NMR (600 MHz, Chloroform-*d*) δ 8.78 (d, *J* = 1.7 Hz, 1H), 7.58 (t, *J* = 2.0 Hz, 1H), 7.41 (d, *J* = 1.9 Hz, 1H), 7.00 (dd, *J* = 15.6, 8.7 Hz, 1H), 5.71 (dd, *J* = 15.6, 3.1 Hz, 1H), 5.31 (dd, *J* = 8.7, 3.1 Hz, 1H), 4.19 (q, *J* = 7.4 Hz, 2H), 1.46 (t, *J* = 7.4 Hz, 3H).

¹³C NMR (151 MHz, Chloroform-*d*) δ 133.54 , 127.88 , 122.80 , 120.75 , 119.51 , 118.62 , 109.96 , 45.51 , 14.68 .

¹⁹F NMR (565 MHz, Chloroform-*d*) δ -79.42.

Figure C2. ¹H-NMR of IL1

Figure C3. ¹³C-NMR of IL1

Figure C4. ^{19}F -NMR of IL1

IL2, Ionic liquid [OMPY][Tf2N]

^1H NMR (600 MHz, Chloroform-*d*) δ 8.82 – 8.72 (m, 2H), 8.48 – 8.42 (m, 1H), 8.00 (t, $J = 7.1$ Hz, 2H), 4.54 (t, $J = 7.6$ Hz, 2H), 1.95 (p, $J = 7.4$ Hz, 2H), 1.33 – 1.14 (m, 10H), 0.80 (t, $J = 7.0$ Hz, 3H).

^{13}C NMR (151 MHz, Chloroform-*d*) δ 145.52 , 144.30 , 128.60 , 118.71 , 62.47 , 31.48 (d, $J = 2.8$ Hz), 28.74 (d, $J = 13.1$ Hz), 25.82 , 22.43 , 13.87 (d, $J = 2.5$ Hz).

^{19}F NMR (565 MHz, Chloroform-*d*) δ -79.13.

Figure C5. ¹H-NMR of IL2

Figure C6. ¹³C-NMR of IL2

Figure C7. ^{19}F -NMR of IL2

We also measured the water contact angle of pristine PEBA membrane (77°), 5% MCM-41-PEBA membrane (65°), 5% MCM-41-IL1-PEBA membrane (85°) and 5% MCM-41-IL2-PEBA membrane (88°). Since molecular sieve MCM-41 has abundant hydrophilic $-\text{OH}$, the water contact angle of the 5% MCM-41-PEBA membrane decreased to 65° , which meant that the hydrophilicity of 5% MCM-41-PEBA membrane was increased. With the incorporation of ionic liquid modified MCM-41, the water contact angles of 5%MCM-41-IL1-PEBA membrane and 5%MCM-41-IL2-PEBA membrane increased to 85° and 88° respectively, which meant that the membranes hydrophobicity was enhanced. Due to IL2 has longer alkyl chain length in its cation, the hydrophobicity of 5%MCM-41-IL2-PEBA membrane was higher than that of 5%MCM-41-IL1-PEBA membrane.

Figure 9. Water contact angle (CA) of different membranes a) Pristine PEBA membrane; b) 5%MCM-41-PEBA membrane; c) 5%MCM-41-IL1-PEBA membrane; d) 5%MCM-41-IL2-PEBA membrane

3) It seems that the authors just load each IL with a certain amount. Did the authors systematically study the IL effect? For example, did the author load more ILs into MCM-41 to get more ILs incorporated into the matrix of MCM-41? The effect of ILs on the butanol pervaporation should be explored.

Response: It's very difficult to accurately measure the ionic content in the mixed matrix membrane. In this manuscript, we did not load more ILs into MCM-41. We have compared the performance of the mixed matrix membranes with different content of ionic liquid modified MCM-41. And the pervaporation experiment data showed that the incorporation of IL1 or IL2 modified MCM-41 is beneficial to butanol separation.

4) What is the mechanism for the enhanced membrane-based separation performance? The authors should perform more experiments and get more better understanding.

Response: MCM-41 is a kind of mesoporous molecular sieve. After IL modification, MCM-41-IL1 and MCM-41-IL2 still retained mesoporous structure. The mesoporous structure of molecular sieve facilitated butanol and water diffusion in the membrane, for which the permeation flux of the modified membrane was significantly improved.

According to the research for butanol extraction by Ha et al [26], the butanol/water selectivity strongly depends on the hydrophobicity of anions of ILs followed by the hydrophobicity of cations of ILs. [Tf2N]⁻ based ionic liquids showed best extraction efficiency for butanol recovery in their research. And the hydrophobicity of cations would increase as the cation alkyl chain length increases. IL2 performed better in selective butanol pervaporation because it has longer alkyl chain length in its cation.

We also measured the water contact angle of pristine PEBA membrane, 5% MCM-41-PEBA membrane, 5% MCM-41-IL1-PEBA membrane and 5% MCM-41-IL2-PEBA membrane. The water contact angle of pristine PEBA membrane was 77°. Because MCM-41 has abundant hydrophilic -OH, the hydrophilicity of 5% MCM-41-PEBA membrane was increased (its contact angle decreased to 65°). However, the contact angles of 5% MCM-41-IL1-PEBA membrane and 5% MCM-41-IL2-PEBA membrane increased to 85° and 88° respectively (5% MCM-41-IL2-PEBA membrane was more hydrophobic). The pervaporation performance of these membranes was consistent

with the water contact angle measurement results.

Reference

26. Ha, S. H., Mai, N. L., Koo, Y.-M. 2010 Butanol recovery from aqueous solution into ionic liquids by liquid–liquid extraction. *Process Biochem.* **45**, 1899–1903. (doi:10.1016/j.procbio.2010.03.030)